# Revisiting VAE for Unsupervised Time Series Anomaly Detection: A Frequency Perspective

## ABSTRACT

Time series Anomaly Detection (AD) plays a crucial role for web systems. Various web systems rely on time series data to monitor and identify anomalies in real time, as well as to initiate diagnosis and remediation procedures. Variational Autoencoders (VAEs) have gained popularity in recent decades due to their superior denoising capabilities, which are useful for anomaly detection. However, our study reveals that VAE-based methods face challenges in capturing long-periodic heterogeneous patterns and detailed short-periodic trends simultaneously. To address these challenges, we propose Frequency-enhanced Conditional Variational Autoencoder (FCVAE), a novel unsupervised AD method for univariate time series. To ensure an accurate AD, FCVAE exploits an innovative approach to concurrently integrate both the global and local frequency features into the condition of Conditional Variational Autoencoder (CVAE) to significantly increase the accuracy of reconstructing the normal data. Together with a carefully designed "target attention" mechanism, our approach allows the model to pick the most useful information from the frequency domain for better short-periodic trend construction. Our FCVAE has been evaluated on public datasets and a large-scale cloud system, and the results demonstrate that it outperforms state-of-the-art methods. This confirms the practical applicability of our approach in addressing the limitations of current VAE-based anomaly detection models.

**Relevence Statement**: Our paper is highly relevant to the track "Internet systems, applications, and Web of Things (WoT) applications." and previous papers in this conference[7, 9, 11, 14, 16, 29, 32, 45, 46]. It presents a novel method for anomaly detection in time series data, which is integral to the monitoring and real-time performance of various web and WoT systems, aligning with the track's focus on web performance, measurements, and characterization. Additionally, our work offers a new perspective on data management and stream processing for web applications, while also sharing experiences and lessons from the deployment of our innovative web-based algorithm.

## CCS CONCEPTS

• **Computing methodologies** → **Machine learning**; • **Security and privacy** → *Systems security*.

*WWW '24, May 13–17, 2024, Singapore*
© 2024 Association for Computing Machinery.
ACM ISBN 978-1-4503-XXXX-X/18/06…$15.00
https://doi.org/XXXXXXX.XXXXXXX

## KEYWORDS

Univariate time series, Anomaly detection, Conditional variational autoencoder, Frequency information

**ACM Reference Format:**
Anonymous Author(s). 2024. Revisiting VAE for Unsupervised Time Series Anomaly Detection: A Frequency Perspective. In *Proceedings of Make sure to enter the correct conference title from your rights confirmation emai (WWW '24)*. ACM, New York, NY, USA, 10 pages. https://doi.org/XXXXXXX. XXXXXXX

## 1 INTRODUCTION

Anomalies are rare in real-world time series data [37], making it difficult to label them and train a supervised model for anomaly detection [46]. Instead, unsupervised machine learning techniques are commonly used [5, 13, 15, 23, 30, 46, 50, 53–55]. These techniques can be divided into two categories: *prediction-based* [13, 15, 55] and *construction-based* [5, 23, 30, 46, 50]. Both types aim to identify normal values and compare them to actual values to detect anomalies. Prediction-based methods were originally developed for forecasting future data points, regardless of whether they were normal or anomalous. However, these methods may overfit to anomalous patterns and underperform. On the other hand, Variational AutoEncoders (VAEs) [18], the leading construction-based approach, encode raw time series into a lower-dimensional latent space and then reconstruct them back to their original dimensions. VAEs are well-suited for detecting anomalies, but existing VAE-based anomaly detection models have not yet reached optimal performance. In this paper, we aim to re-examine the VAE model and improve its effectiveness in anomaly detection.

In order to more effectively demonstrate the challenges associated with VAE-based techniques, we provide an example in Figure 1. The original curve is displayed in the first sub-figure, with anomalies highlighted in red ③. The subsequent four sub-figures represent curves reconstructed by four distinct VAE-methods, including our proposed method (referred to as FCVAE). The reconstruction error is indicated by the green shaded area ⑤. To achieve superior AD performance, the reconstructed result should closely resemble the original curve for normal points, while deviating significantly for anomalous points ③. As evident in the figure, all VAE-based methods successfully disregard the anomalies during reconstruction. However, the reconstruction results for some normal points, particularly those marked by a blue rectangle ① and ellipse ④, are not satisfactory. This substantially impacts the overall performance, leading us to identify three key challenges that we address in the subsequent sections.

**Challenge 1: Capturing similar yet heterogeneous periodic patterns.** From Figure 1, periodic patterns can be observed in the curves, with one such period emphasized by the red shaded area ②. However, the shapes across different periods vary. As demonstrated by the blue ellipse, existing VAE-based methods (as shown

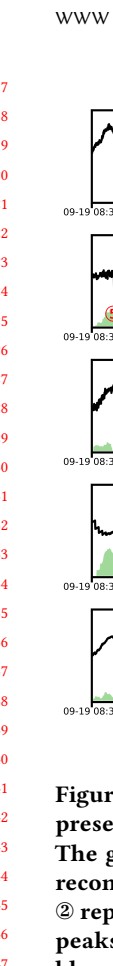

**Figure 1: Comparison of four KPI reconstruction methods presented in our paper, highlighting anomalies in red ③. The green shade ⑤ represents the difference between the reconstructed values and the original values, the red shade ② represents a long period, and the blue ellipse ④ indicates peaks and valleys that are not properly reconstructed, the blue rectangle ① will be magnified in Figure 2 for detailed comparison.**

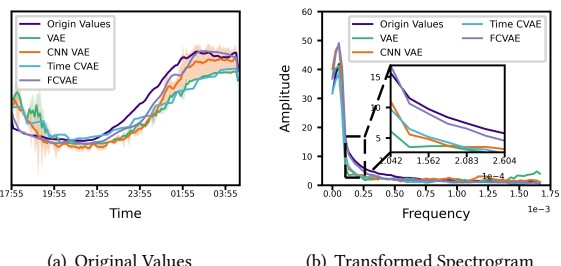

**Figure 2: A detailed view of the region enclosed by a blue rectangle ① in Figure 1, where the shaded area represents the value range before applying a sliding window average.**

in the second sub-figure) are unable to capture these heterogeneous patterns effectively. This observation naturally leads to the idea of utilizing conditional VAE to map data into distinct Gaussian spaces by considering the timestamp as a condition. Unfortunately, as illustrated in the fourth sub-figure (Time CVAE [23]), the results are unsatisfactory, which we will further discuss below.

**Challenge 2: Capturing detailed trends.** Reconstructing monotonous patterns (*i.e.,* trends) might appear straightforward at first glance. However, upon a closer examination of the local window (highlighted in a blue rectangle ① in Figure 1 and magnified in Figure 2(a), it becomes evident that existing methods fail to restore detailed patterns within this time frame. In Figure 2(a), the two green lines initially overestimate the ground truth (purple curve) but subsequently underestimate it for the remainder of the window. This is primarily because existing methods aim to minimize the overall reconstruction error without focusing on "point-to-point" dependencies, *e.g.,* the precise upward and downward ranges following a specific point. This omission results in fluctuating reconstruction outcomes (as seen in the second sub-figure). Although CNN attempts to model point-to-point dependencies within the window, it still produces coarse-grained fluctuations (visible in the third sub-figure in Figure 1). The reason of unsatisfactory result of CNN-CVAE lies in Figure 2(b). Upon converting the curves reconstructed by various methods (Figure 2(b)), it becomes evident that the primary cause of these phenomena is the **absence of some frequencies** (smaller amplitude of certain frequencies) in existing methods, hindering the reconstruction of detailed patterns. This observation logically suggests the possibility of employing frequency as the conditional factor in a Conditional Variational Autoencoder (CVAE). Nonetheless, employing frequency as the condition in CVAE presented a new challenge.

**Challenge 3: A large number of sub-frequencies make the signal in condition noisy and difficult to use.** Directly converting the entire window into the frequency-domain results in numerous sub-frequencies, adding noise and obstructing effective VAE-based reconstruction. To address these challenges, we sub-divide the entire window into smaller ones and propose a **target attention** method to select the most useful sub-window frequencies.

In this paper, we introduce a novel unsupervised anomaly detection algorithm, named FCVAE (Frequency-enhanced Conditional Variational AutoEncoder). Different from current VAE-based anomaly detection methods, FCVAE innovatively incorporates both global and local frequency information to guide the encoding-decoding procedures, that both heterogeneous periodic and detailed trend patterns can be effectively captured. This in turn enables more accurate anomaly detection. Our paper's contributions can be summarized as follows:

- Our analysis of the widely-used VAE model for anomaly detection reveals that existing VAE-based models fail to capture both heterogeneous periodic patterns and detailed trend patterns. We attribute this failure to the missing of some frequency-domain information, which current methods fail to reconstruct.
- Our study systematically improves the long-standing VAE by focusing on frequency. Our proposed FCVAE makes the VAE-based approach the state-of-the-art in anomaly detection once more. This is significant because VAE-based methods can inherently handle mixed anomaly-normal training data, while prediction-based methods cannot.
- Evaluations demonstrate that our FCVAE substantially surpasses state-of-the-art methods (0%–40% on public datasets and 10% in **a real-world web system** in terms of F1 score).

Comprehensive ablation studies provide an in-depth analysis of the model, revealing the reasons behind its superior performance.

The replication package for this paper, including all our data, source code, and documentation, is publicly available online at **https://anonymous.4open.science/r/FCVAE**.

## 2 PRELIMINARIES

### 2.1 Problem Statement

To facilitate comprehension, we employ the notation established by [22]. Given a UTS data $\mathbf{x} = [x_0, x_1, x_2, \cdots, x_t]$ and label series $\mathbf{L} = [l_0, l_1, l_2, \cdots, l_t]$, where $x_i \in \mathbb{R}$, $l_i \in \{0, 1\}$, and $t \in \mathbb{N}$. $\mathbf{x}$ represents the entire time series data array, while $x_i$ signifies the metric value at time $i$. $\mathbf{L}$ denotes the label of time series $\mathbf{x}$. We define the UTS anomaly detection task as follows:

*Given a UTS* $\mathbf{x} = [x_0, x_1, x_2, \cdots, x_t]$, *the objective of UTS anomaly detection is to utilize the data* $[x_0, x_1, \cdots, x_{i-1}]$ *preceding each point* $x_i$ *to predict* $l_i$. *Based on the value of* $l_i$, *we can determine whether* $x_i$ *is an anomaly or not.*

### 2.2 VAEs and CVAEs

VAE is composed of an encoder $q_\phi(\mathbf{z}|\mathbf{x})$ and a decoder $p_\theta(\mathbf{z}|\mathbf{x})$. VAE can be trained by using the reparameterization trick. SGVB [35] is a commonly used training method for VAE because of its simplicity and effectiveness. It maximizes the evidence lower bound (ELBO) to simultaneously train the reconstruction and generation capabilities of VAE. The ELBO is defined in (1).

$$\mathcal{L} = \mathbb{E}_{q_\phi(\mathbf{z}|\mathbf{x})}\left[\log p_\theta(\mathbf{x}|\mathbf{z}) + \log p_\theta(\mathbf{z}) - \log q_\phi(\mathbf{z}|\mathbf{x})\right] \quad (1)$$

DONUT [46] proposed the modified ELBO (M-ELBO) to weaken the impact of abnormal and missing data in the window on the reconstruction. M-ELBO is defined in (2), $\alpha_w$ is defined as an indicator, where $\alpha_w = 1$ indicates $x_w$ being not anomalous or missing, and $\alpha_w = 0$ otherwise. $\beta$ is defined as $(\sum_{w=1}^{W} \alpha_w)/W$.

$$\mathcal{L} = \mathbb{E}_{q_\phi(\mathbf{z}|\mathbf{x})}\left[\sum_{w=1}^{W} \alpha_w \log p_\theta(x_w|\mathbf{z}) + \beta \log p_\theta(\mathbf{z}|\mathbf{x}) - \log q_\phi(\mathbf{z}|\mathbf{x})\right] \quad (2)$$

The overall structure of CVAE [39] is similar to VAE, and it combines conditional generative models with VAE to achieve stronger control over the generated data. The training objective of CVAE is defined as (3), where $\mathbf{c}$ is the condition, similar to that of VAE.

$$\mathcal{L} = \mathbb{E}_{q_\phi(\mathbf{z}|\mathbf{x},\mathbf{c})}\left[\log p_\theta(\mathbf{x}|\mathbf{z}, \mathbf{c}) + \log p_\theta(\mathbf{z}) - \log q_\phi(\mathbf{z}|\mathbf{x}, \mathbf{c})\right] \quad (3)$$

## 3 METHODOLOGY

### 3.1 Framework Overview

The proposed algorithm for anomaly detection is illustrated in Figure 3 and comprises three main components: data preprocessing, training, and testing. Given a data point $x_t$, as defined in the problem statement, its state can only be evaluated based on its preceding points $[x_0, x_1, \cdots, x_t]$. To maintain model consistency, a sliding window method is employed, where a window of $W$ consecutive points, $[x_{t-W+1}, x_{t-W+2}, \cdots, x_t]$, is utilized to determine if $x_t$ is anomalous. Following sliding window and data preprocessing, a batch of data is input into the FCVAE model for offline training, which will be presented in detail later. Subsequently, the trained

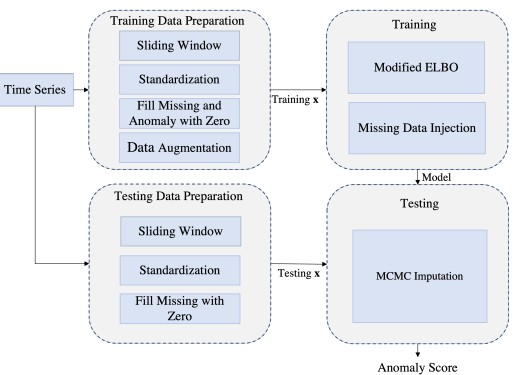

**Figure 3: Overall Framework.**

model is transferred to the online test module for testing and computing the anomaly score.

### 3.2 Data Preprocessing

Data preprocessing encompasses standardization, filling missing and anomaly points, and the newly introduced method of **data augmentation**. The efficacy of data standardization and filling missing and anomaly points has been substantiated in prior studies [23, 26, 46]. Therefore, we directly incorporate these techniques into our approach.

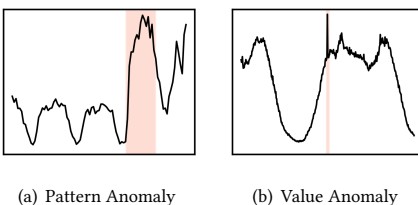

(a) Pattern Anomaly      (b) Value Anomaly

**Figure 4: Examples of the two most frequent anomalies, where the red shaded area denotes the abnormal segments.**

Previous data augmentation methods [21, 43, 49] often added normal samples, such as variations of data from the time domain or frequency domain. However, for our method, we train the model by incorporating all the time series from the dataset together, which provides sufficient pattern diversity. Furthermore, FCVAE has the ability to extract pattern information due to the addition of frequency information, so it can handle new patterns well. Nonetheless, even with the introduction of frequency information, anomalies are often challenging to be effectively addressed. For the model to learn how to handle anomalies, we primarily focus on abnormal data augmentation. In time series data, anomalies are mostly manifested as pattern mutations or value mutations (shown in Figure 4), so our data augmentation mainly targets to these two aspects. The augmentation on the pattern mutation is generated by combining two windows from different curves, with the junction acting as the anomaly. Value mutation refers to changing some points in the window to randomly assigned abnormal values. With the augmented

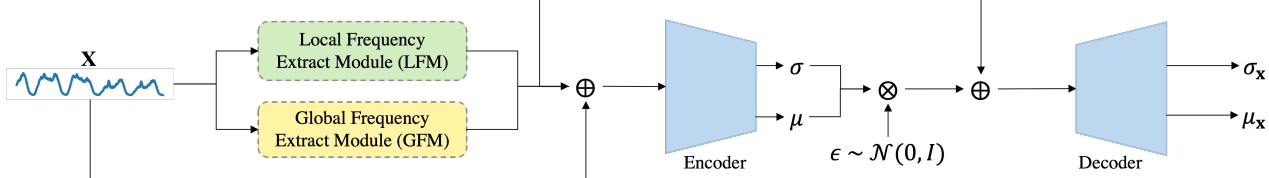

**Figure 5: FCVAE Model Architecture.**

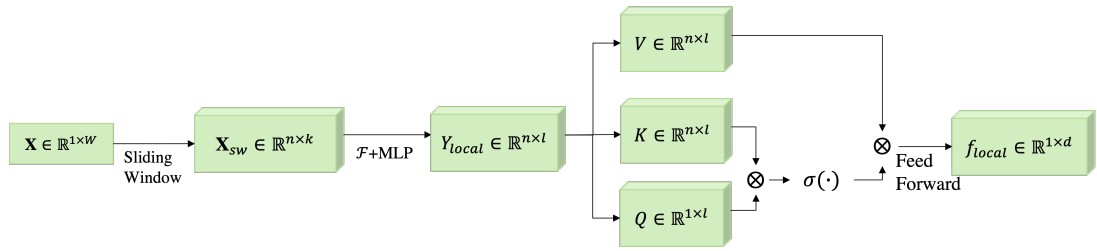

**Figure 6: Architecure of LFM.**

anomaly data, M-ELBO in CVAE, which will be introduced in detail later, can perform well even in an unsupervised setting without true labels.

## 3.3 Network Architecture

The proposed FCVAE model is illustrated in Figure 5. It comprises three main components: encoder, decoder, and a condition extraction block that includes a global frequency information extraction module (GFM) and a local frequency information extraction module (LFM). Equation (4) illustrates how our model works.

$$\mu, \sigma = \text{Encoder}(\mathbf{x}, \text{LFM}(\mathbf{x}), \text{GFM}(\mathbf{x}))$$
$$\mathbf{z} = \text{Sample}(\mu, \sigma) \quad (4)$$
$$\mu_{\mathbf{x}}, \sigma_{\mathbf{x}} = \text{Decoder}(\mathbf{z}, \text{LFM}(\mathbf{x}), \text{GFM}(\mathbf{x}))$$

*3.3.1 GFM.* The GFM module (Figure 7) extracts the global frequency information using the FFT transformation ($\mathcal{F}$). However, not all frequency information is useful. The frequencies resulted from the noise and anomalies in the time series data appear as long tails in the frequency domain. Therefore, we employ a linear layer after the FFT to filter out the useful frequency information that can represent the current window pattern. Moreover, we incorporate a dropout layer following Fedformer [56] to enhance the model's ability to learn the missing frequency information.

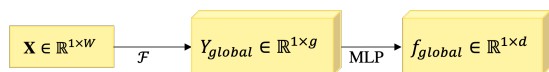

**Figure 7: Architecure of GFM.**

The $f_{global} \in \mathbb{R}^{1 \times d}$ is calculated as (5), where d is the embedding dimension of the global frequency information and $\mathcal{F}$ means FFT.

$$f_{global} = \text{Dropout}(\text{Dense}(\mathcal{F}(\mathbf{x}))) \quad (5)$$

*3.3.2 LFM.* The attention mechanism [42] has been widely adopted in time series data processing due to its ability to dynamically process dependencies between different time steps and focus on important ones. Target attention, which is developed based on attention, is widely used in the field of recommendation [4]. Specifically, target attention can weigh the features of the target domain, leading to more accurate domain adaptation.

The GFM module extracts the frequency information from the entire window, proving to be effective in reconstructing the data within the whole window. However, we use a window to detect whether the last point is abnormal, which poses a challenge because the GFM module does not provide sufficient attention to the last point. This can result in a situation where the reconstruction is satisfactory for part of the window but not for another part, especially when changes in system services lead to the concept drift in the time series data. Even in the absence of concept drift, GFM cannot capture local changes as it extracts the average frequency information from the entire window; hence, the reconstruction of the last key point may be unsatisfactory. Nonetheless, as previously mentioned, target attention can effectively address this issue, as it captures the frequency information of the entire window while paying a greater attention to the latest time point. Therefore, we propose the LFM that incorporates the target attention.

As depicted in Figure 6, the LFM module operates by sliding the entire window $\mathbf{x}$ to obtain several small windows $\mathbf{x}_{sw}$. Subsequently, FFT and frequency information extraction are applied to each small window. The most recent small window is used as the query $Q$ because it contains the last point that we want to detect. The remaining small windows are utilized as keys $K$ and values $V$ for target attention. Finally, a linear layer is employed to facilitate the model in learning to extract the most important and useful part of the local frequency information, and dropout is also applied to enhance the model's ability to reconstruct some of the local frequency information like GFM.

 

$$\mathbf{x}_{sw} = \text{SlidingWindow}(\mathbf{x})$$
$$Q = \text{Select}(\text{Dense}(\mathcal{F}(\mathbf{x}_{sw})))$$
$$K, V = \text{Dense}(\mathcal{F}(\mathbf{x}_{sw})) \tag{6}$$
$$f_{local} = \text{Dropout}(\text{FeedFawrd}((\sigma(Q \cdot K^\top) \cdot V))$$

The calculation of $f_{local} \in \mathbb{R}^{1 \times d}$ in LFM is given by (6), where $d$ is the embedding dimension of the local frequency information, which is the same as that of GFM. Here, $\mathbf{x}_{sw} \in \mathbb{R}^{n \times k}$ represents a group of small windows extracted from the original window, where $k$ is the dimension of the small windows and $n$ is the number of small windows. The Select function is employed to select the latest window as the query $Q$ and the Dense function means dense neural network. The softmax function $\sigma$ is used to calculate the attention weights for the small windows.

## 3.4 Training and Testing

The training process of FCVAE incorporates three key technologies: CVAE-based modified evidence lower bound (CM-ELBO), missing data injection, as well as the newly proposed **masking the last point**. As shown in (7), **CM-ELBO** is obtained by applying M-ELBO [46] to CVAE. Missing data injection [26, 46] is a commonly used technique in VAE that we directly apply. We observed that an anomalous point in time series data manifests as an outlier value in the time domain. However, when the data are transformed into the frequency domain, all frequency information is shifted, leading to a challenge. The impact of this issue will be amplified when the last point is abnormal, as we specifically aim to detect whether the last point is abnormal given the whole window. While we use the frequency enhancement method and frequency selection to mitigate this problem to some extent, we mask the last point as zero during the extraction of the frequency condition to address this issue further.

$$\mathcal{L} = \mathbb{E}_{q_\phi(\mathbf{z}|\mathbf{x},\mathbf{c})} \left[ \sum_{w=1}^{W} \alpha_w \log p_\theta(x_w|\mathbf{z},\mathbf{c}) + \beta \log p_\theta(\mathbf{z}) - \log q_\phi(\mathbf{z}|\mathbf{x},\mathbf{c}) \right] \tag{7}$$

While testing, FCVAE adopts the Markov Chain Monte Carlo (MCMC)-based missing imputation algorithm proposed in [35] and applied in [26] to mitigate the impact of missing data. Since our goal is to detect the last point of a window, the last point is set to missing for MCMC to obtain a normal value. This also allows for a better adaptation to the last point mask mentioned earlier. FCVAE further utilizes reconstruction probabilities as anomaly scores, which are defined in equation (8).

$$\text{AnomalyScore} = -\mathbb{E}_{q_\phi(\mathbf{z}|\mathbf{x},\mathbf{c})} \left[ \log p_\theta(\mathbf{x}|\mathbf{z},\mathbf{c}) \right] \tag{8}$$

## 4 EXPERIMENTS

### 4.1 Experiment Settings

*4.1.1 Datasets.* To evaluate the effectiveness of our proposed algorithm, we conducted experiments on four datasets. **Yahoo** [2] is an open data set for anomaly detection released by Yahoo lab. **KPI** [24] KPI is collected from five large Internet companies (Sougo, eBay, Baidu, Tencent, and Ali). **WSD** [1] Web service dataset (WSD) contains real-world KPIs collected from three top-tier Internet companies, Baidu, Sogou, and eBay, providing large-scale Web services. **NAB** [20] The Numenta Anomaly Benchmark (NAB) is an open

dataset created by Numenta company for evaluating the performance of time series anomaly detection algorithms.

*4.1.2 Baseline Methods.* To benchmark our model FCVAE against existing methods, we chose the following approaches for evaluation: SPOT [38], SRCNN [34], TFAD [49], DONUT [46], Informer [55], Anomaly-Transformer [47], AnoTransfer [50], VQRAE [17]. SPOT represents a traditional statistical method rooted in extreme value theory. SRCNN and TFAD are supervised methods relying on high-quality labels. Donut, VQRAE, and AnoTransfer are unsupervised reconstruction-based methods utilizing VAE for normal value reconstruction. Informer is an unsupervised prediction-based method that endeavors to predict normal values using an attention mechanism. Anomaly-Transformer is an unsupervised anomaly detection method leveraging the transformer architecture. It posits that normal data exhibit stronger correlations with distant data.

*4.1.3 Evaluation Metrics.* In practical applications, operators tend to be less concerned with point-wise anomaly detection, i.e., whether each individual point is classified as anomalous or not, and focus more on detecting continuous anomalous segments in time series data. Moreover, due to the substantial impact of anomalous segments, operators aim to identify such segments as early as possible. To address these requirements, we adopt two metrics, best F1 and delay F1, which are based on the works of DONUT [46] and SRCNN [34], respectively.

| truth | 0 | 0 | 1 | 1 | 1 | 0 | 1 | 1 | 1 | 1 |
|---|---|---|---|---|---|---|---|---|---|---|
| predict | 1 | 0 | 0 | 1 | 1 | 0 | 0 | 0 | 1 | 1 |
| adjusted | 1 | 0 | 1 | 1 | 1 | 0 | 1 | 1 | 1 | 1 |
| delay adjusted | 1 | 0 | 1 | 1 | 1 | 0 | 0 | 0 | 0 | 0 |

**Figure 8: Illustration of the adjustment strategy.**

Best F1 is obtained by traversing all possible thresholds for anomaly scores, and subsequently applying a point adjustment strategy to the prediction in order to compute the F1 score. Delay F1 is similar to best F1 but employs a delay point adjustment strategy to transform the prediction. The adjustment strategies are illustrated in Figure 8, with a delay set to 1 as an example. The detector misses the second anomalous segment because it takes two-time intervals to detect this segment, exceeding the maximum delay threshold we established. We configure the delay for all datasets to be 7, except for Yahoo, where it is set to 3, and NAB, where it is set to 150. This is because the anomaly segments in Yahoo are very short, while in NAB, they are typically much longer, often spanning several hundred data points.

*4.1.4 Implementation Details.* To guarantee the widespread applicability, all the experiments described below were conducted under entirely unsupervised conditions, without employing any actual labels (all labels are set to zero). For consistency across all methods, we trained a single model for all curves within a dataset. Regarding hyperparameters, we conducted a grid search to identify the most effective parameters for different datasets. Additionally, we later evaluated the sensitivity of these parameters to ensure robust performance.

**Table 1: Performance on test data.**

| Method | Yahoo | | KPI | | WSD | | NAB | |
|---|---|---|---|---|---|---|---|---|
| | Best F1 | Delay F1 | Best F1 | Delay F1 | Best F1 | Delay F1 | Best F1 | Delay F1 |
| **SPOT** [38] | 0.417 | 0.417 | 0.360 | 0.143 | 0.472 | 0.237 | 0.829 | 0.829 |
| **SRCNN** [34] | 0.251 | 0.198 | 0.786 | 0.678 | 0.170 | 0.053 | 0.828 | 0.575 |
| **DONUT** [46] | 0.215 | 0.215 | 0.454 | 0.364 | 0.224 | 0.158 | 0.935 | 0.797 |
| **VQRAE** [17] | 0.510 | 0.492 | 0.272 | 0.137 | 0.312 | 0.103 | 0.933 | 0.893 |
| **Anotransfer** [50] | 0.567 | 0.496 | 0.685 | 0.461 | 0.674 | 0.379 | 0.965 | 0.871 |
| **Informer** [55] | 0.707 | 0.671 | **0.918** | **0.822** | 0.557 | 0.393 | **0.973** | 0.892 |
| **TFAD** [49] | **0.805** | **0.802** | 0.752 | 0.680 | 0.628 | **0.455** | 0.734 | 0.248 |
| **Anomaly-Transformer** [47] | 0.274 | 0.029 | 0.868 | 0.346 | **0.728** | 0.137 | 0.971 | **0.911** |
| **FCVAE** | **0.857** | **0.842** | **0.927** | **0.835** | **0.831** | **0.631** | **0.976** | **0.917** |

## 4.2 Overall Performance

The performance of FCVAE and baseline methods across the four datasets is depicted in Table 1. Our method surpasses all baselines on four datasets regarding best F1 by 6.45%, 0.98%, 14.14% and 0.31%. In terms of delay F1, our method outperforms all baselines on four datasets by 4.98%, 1.58%, 38.68% and 0.65%.

The performance of various baseline methods on the datasets exhibits considerable variation. For instance, SPOT [38] does not excel on most datasets, as it erroneously treats extreme values as anomalies, whereas anomalies are not always manifested as such. SRCNN [34] is a reasonably proficient classifier, yet its performance falls short compared to most other models. This underscores the fact that implicitly extracting abnormal features is challenging. Informer [55] outperforms most other baselines across different datasets, as many anomalies exhibit notable value jumps, and prediction-based methods can effectively manage this situation. However, it struggles with anomalies induced by frequency changes. Anomaly-Transformer [47] attains commendable results on most datasets in terms of best F1 but demonstrates a low delay F1. It detects anomalies based on the relationships with nearby points, and only when the anomalous point is relatively central within the window can it easily capture the correlation. Conversely, TFAD [49] achieves favorable results on various datasets but exhibits a certain delay in detection.

Moreover, our method surpasses DONUT [46] and VQRAE [17] in terms of reconstruction-based methods. Although VQRAE [17] introduces numerous modifications to the VAE, employing RNN to capture temporal relationships, our method still outperforms it. This finding implies that for UTS anomaly detection, it is imperative to incorporate only key information while avoiding overloading the model with superfluous data.

## 4.3 Different Types of Conditions in CVAE

In this context, we conduct experiments under identical settings to evaluate different types of conditions. The chosen conditions encompass information potentially useful for time series anomaly detection within the scope of our understanding, including timestamps [50], time domain information, and frequency domain information. To ensure consistency, we apply the same operation on the time domain information as we do on the frequency domain information.

As illustrated in Figure 9(a), the performance of employing the frequency information as a condition surpasses that using the timestamp or time domain information. This can be readily comprehended, as timestamps carry limited information and typically require one-hot encoding, resulting in sparse data representation. Time domain information is already incorporated in VAE, and utilizing it as a condition may lead to redundant information without significantly benefiting the reconstruction. Conversely, **frequency information, as a valuable and complementary prior, rendering it a more effective condition for anomaly detection.**

## 4.4 Frequency VAE and FACVAE

Is CVAE the optimal strategy for harnessing the frequency information in anomaly detection? In this study, we compare FCVAE with an improved frequency-based VAE (FVAE) model, in which the frequency information is integrated into VAE along with the input to reconstruct the original time series. As depicted in Figure 9(b), FCVAE surpasses FVAE. This outcome can be attributed to two primary reasons. Firstly, CVAE, due to its unique architecture that incorporates conditional information, intrinsically outperforms VAE in numerous applications. Secondly, FVAE does not fully exploit frequency information. Although it incorporates this additional information, it still lacks efficient utilization in practice, particularly in the decoder. Consequently, **the CVAE that incorporates the frequency information as a condition represents the most effective structure known to date.**

## 4.5 GFM and LFM

We propose GFM and LFM to extract global and local frequency information, respectively. However, do these two modules achieve our intended effects through their designs? Additionally, it is worth noting that GFM and LFM may overlap to some degree. Thus, we would like to determine if combining the two can further enhance the performance.

We conduct experiments and the results are depicted in Figure 9(c). It can be observed that, across the four datasets, employing either LFM or GFM in FCVAE outperforms the VAE model under

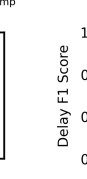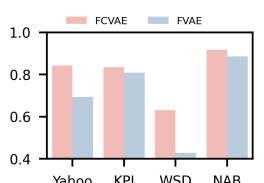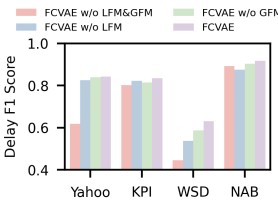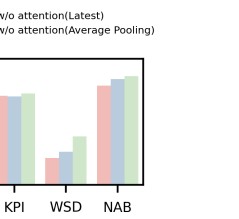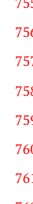

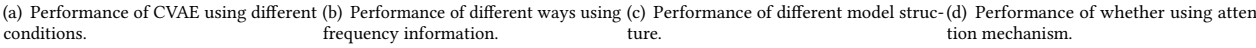

(a) Performance of CVAE using different conditions.

(b) Performance of different ways using frequency information.

(c) Performance of different model structure.

(d) Performance of whether using attention mechanism.

**Figure 9: Delay F1 score of different settings.**

identical conditions of other settings except for NAB, where the frequent oscillation of data results in inconsistency between the information extracted from GFM and the data value of the current time. For all datasets, when both LFM and GFM modules are utilized concurrently, they synergistically enhance each other, resulting in superior performance. Consequently, **both global and local frequency information play a crucial role in detecting anomalies.**

### 4.6 Attention Mechanism

It is crucial to discern whether the enhancement in LFM stems from the reduced window size or the attention mechanism. Thus, we perform experiments by excluding the attention operation from LFM while keeping GFM unaltered. Specifically, we utilized frequency information either from the latest small window in LFM (Latest) or from the average pooling of frequency information across all small windows in LFM (Average Pooling).

The findings in Figure 9(d) demonstrate that without attention, it is impossible to attain the original performance of FCVAE since it is not feasible to determine the specific weight of each small window in advance. However, **the attention mechanism effectively addresses this issue by assigning higher weights to more informative windows.**

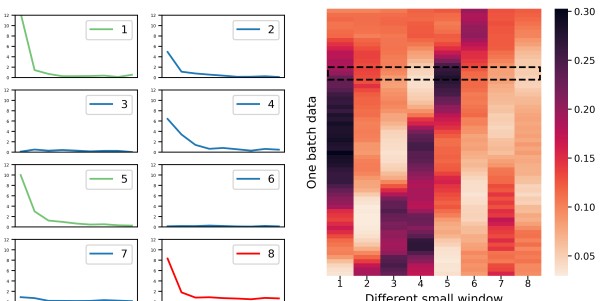

(a) Spectrum of small windows for data in the black dashed box on the right.

(b) Heatmap of LFM attention in a batch. The 8-th window is the latest window.

**Figure 10: A example of how attention mechanism works in LFM.**

We present a comprehensive explanation of the attention mechanism in LFM using a case. A specific data segment, denoted by the black dashed box in Figure 10(b), is selected and all small windows produced by LFM's sliding window module are transformed into the frequency domain to obtain their spectra. As illustrated in Figure 10(a), the 5-th (green) and the 8-th (red) windows exhibit the highest similarity, where the 8-th window serves as the query $Q$ for our attention. Upon examining Figure 10(b), it can be observed that the heat value of the 5-th window is the highest, which corresponds with the findings in Figure 10(a). Furthermore, we observe from Figure 10(b) that the weight changes of LFM attention have a certain temporal gradient. This is easily understood since adjacent windows are similar.

### 4.7 Key Techniques in Framework

In this section, we evaluate the effectiveness of our novel data augmentation technique, masking the last point, and the application of CM-ELBO on four distinct datasets. The results are presented in Table 2. Based on the results, it is clear that CM-ELBO plays the most crucial role in most datasets, which aligns with our expectations. This is because it can tolerate abnormal or missing data to a certain extent. Furthermore, masking the last point has a substantial impact on the results, as when an anomaly occurs at the last point of the window, it affects the entire frequency information. Effectively masking this point resolves the issue and improves the detection accuracy. Data augmentation, on the other hand, introduces some artificial anomalies to boost the performance of CM-ELBO, particularly in unsupervised settings.

**Table 2: Delay F1 of different settings.**

| Variants | Yahoo | KPI | WSD | NAB |
|---|---|---|---|---|
| w/o data augment | 0.841 | 0.825 | 0.626 | 0.904 |
| w/o mask last point | 0.835 | 0.830 | 0.534 | 0.877 |
| w/o CM-ELBO | 0.690 | 0.757 | 0.435 | 0.897 |
| **FCVAE** | **0.842** | **0.835** | **0.631** | **0.917** |

### 4.8 Parameter Sensitivity

Our architecture incorporates several techniques, each with its own set of parameters. The stability of a model to different parameters

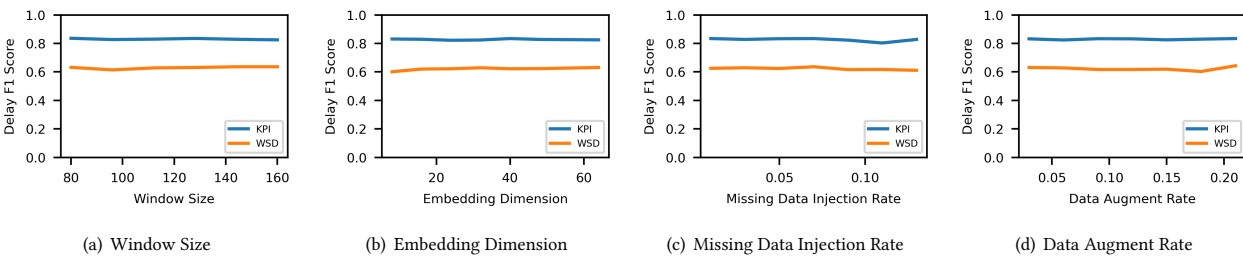

Figure 11: Delay F1 score of different settings.

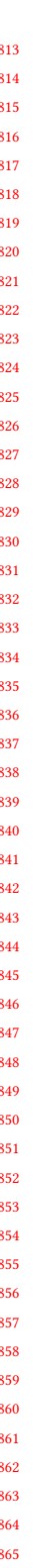

is an important aspect to consider, and therefore we test the sensitivity of our model parameters on two datasets, KPI and WSD. We examine four aspects: the dimension of the condition, the window size, the proportion of missing data injection, and the proportion of data augmentation. The results, shown in Figure 11, indicate that our model can achieve stable and excellent results under different parameter settings.

## 5 PRODUCTION IMPACT AND EFFICIENCY

Our FCVAE approach has been incorporated as a crucial component in a large-scale cloud system that caters to millions of users globally. The system generates billions of time series data points on a daily basis. The FCVAE detects anomalies in the cloud system, with the primary goal of identifying any potential regressions in the system that may indicate the occurrence of an incident.

**Table 3: Online performance of FCVAE in production compared to legacy detector. F1 represents Best F1, F1* represents Delay F1.**

| Baseline | | FCVAE | | Improvement | | Inference efficiency |
|---|---|---|---|---|---|---|
| F1 | F1* | F1 | F1* | F1 | F1* | [points/second] |
| 0.66 | 0.63 | 0.73 | 0.69 | 10.9% | 11.1% | 1195.7 |

Table 3 presents the online performance improvement achieved by employing FCVAE over a period of one year. The experiments were conducted on a 24GB memory 3090 GPU. The results demonstrate substantial enhancements in both Best F1 and Delay F1 compared to the legacy detector. This underscores the effectiveness and robustness of our proposed method. Furthermore, our model is lightweight and highly efficient, capable of processing over 1000 data points within 1 second. This far exceeds the speed at which the system generates new temporal points.

## 6 RELATED WORK

**Traditional statistical methods** [28, 31, 36, 41, 52] are widely used in time series anomaly detection because of their great advantages in time series data processing. For example, [33] find the high frequency abnormal part of data through FFT [41] and verify it twice. Twitter [40] uses STL [6] to detect anomaly points. SPOT [38] considers that some extreme values are abnormal, therefore, detects them through Extreme Value Theory [8].

**Supervised methods** [19, 27, 34] mostly learn the features of anomalies and identify them through classifiers based on the features learned. Opprentice [27] efficiently combines the results of many detectors through random forest. SRCNN [34] build a classifier through spectral residual [12] and CNN. Some methods [3, 49] obtain pseudo-labels through data augmentation to enhance the learning ability.

**Unsupervised methods** are mainly divided into reconstruction-based and prediction-based methods. Reconstruction-based methods [5, 17, 23, 25, 46] learn low-dimensional representations and reconstruct the "normal patterns" of data and detect anomalies according to reconstruction error. DONUT [46] proposed the modified ELBO to enhance the capability of VAE. Buzz [5] is the first to propose a deep generative model. ACVAE [25] adds active learning and contrastive learning. Prediction-based methods [15, 55] try to predict the normal values of metrics based on historical data and detect anomalies according to the prediction error. LSTM [15] proposes to use the LSTM model to predict normal values. Informer [55] changes the relevant mechanism of self attention. In recent years, transformer-based methods have been widely proposed. Anomaly-Transformer [47] detect anomalies by comparing Kullback-Leible (KL) divergence between two distributions. In recent years, some methods [44, 48, 56] have begun to solve some practical problems from the frequency domain. Moerover, many transfer learning methods have been proposed in recent years [10, 23, 50, 51] since they are very fast.

## 7 CONCLUSION

Our paper presents a novel unsupervised method for detecting anomalies in UTS, termed FCVAE. At the model level, we introduce the frequency domain information as a condition to work with CVAE. To capture the frequency information more accurately, we propose utilizing both GFM and LFM to concurrently capture the features from global and local frequency domains, and employing the target attention to more effectively extract the local information. At the architecture level, we propose several new key technologies, including CM-ELBO, the augmentation of anomalous data and masking the last point when extracting the frequency information during training. These innovations further improves the detection accuracy and efficiency. We carry out experiments on four dataset and an online cloud system to evaluate our approach's accuracy, and comprehensive ablation experiments to demonstrate the effectiveness of each module.

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
