# OpenReview forum: "Revisiting VAE for Unsupervised Time Series Anomaly Detection: A Frequency Perspective"
_ACM.org/TheWebConf/2024/Conference — TheWebConf24_

### Official Review · Reviewer_N23X · 2023-11-19

**Novelty:** 4
**Technical Quality:** 4

**Review:**

This paper presents a Conditional Variational Autoencoder-based anomaly detection method that incorporates frequency information.

**Pros:**

1. The paper is well-organized, featuring high-quality figures.
2. The challenge 1 is an important and interesting problem, which refers to detecting anomaly by a unified model from data with diverse normal patterns.

**Cons:**

1. The challenges proposed in this paper lack a comprehensive assessment of current anomaly detection methods. For instance, Challenge 2 asserts that Variational Autoencoders (VAEs) struggle with capturing detailed trends because they focus on minimizing overall reconstruction error instead of point-to-point dependencies. However, existing works, such as OmniAnomaly[1], fuse VAE with recurrent networks, considering temporal dependencies and effectively handling detailed trends.

   [1] Su Y, Zhao Y, Niu C, et al. Robust anomaly detection for multivariate time series through stochastic recurrent neural network. Proceedings of the 25th ACM SIGKDD international conference on knowledge discovery & data mining. 2019: 2828-2837.

2. The motivation behind introducing frequency information is unclear. The authors claim that VAE-based models fail to capture both heterogeneous periodic patterns and detailed trend patterns due to missing information in the frequency domain (lines 217-222). However, the connection between capturing heterogeneous periodic patterns and utilizing frequency domain information is not self-evident and requires further clarification.

3. It is difficult to say that the proposed method is significantly novel, as it merely combines Fedformer, MLP, attention mechanisms, and CVAE.

4. The proposed method only demonstrates a strong advantage over baselines on one dataset (WSD), while showing only marginal improvement on other datasets, especially KPI and NAB.

5. Some parts of the paper are hard to pass. For instance, non-standard terms are used without clarification (e.g., "sub-frequencies" in line 200). Additionally, there are grammar issues (e.g., 'A large number of sub-frequencies make the signal in condition noisy and difficult to use.' What does 'make the signal in condition noisy' mean?)."

**Questions:**

Please refer to cons in review.

**Ethics Review Description:**

No ethics issue.

**Reviewer Confidence:**

3: The reviewer is confident but not certain that the evaluation is correct

**Scope:**

4: The work is relevant to the Web and to the track, and is of broad interest to the community

---

### Official Review · Reviewer_4SxQ · 2023-11-20

**Novelty:** 4
**Technical Quality:** 4

**Review:**

The authors present a solution to identify anomalies in univariate Time Series. They propose to enhance the performance of VAEs (Variational Autoencoders) by combining global and local features into an autoencoder.


While authors and to the paper a "Relevant Statement" paragraph to justify the fit of the paper into the "Systems and Infrastructure for Web, Mobile, and WoT”, in my opinion the paper is clearly out of the scope of this track, and it should be rejected for this reason.
The authors provide three arguments to justify the fit of the paper into this track:

 1 - Web and WoT systems use anomaly detection in time series data for the monitoring of system performance. This might be true, but this does not justify the paper being a fit to the track. Web systems are currently deployed in Cloud Providers (e.g., AWS, Google Cloud, etc.) but an improvement in the performance of deployment of applications in a cloud system, wouldn't not make a paper in that topic to fit in the current track. Likewise, web system performance depends on the network infrastructure, hence a paper that propose an improvement in the network infrastructure or network protocols (e.g., a new version of tcp which makes web systems faster) wouldn't be a fit for this track, despite a web system may benefit from using the new version of the TCP protocol.


2- The paper offers a new perspective on data management and stream processing for web applications, while also sharing experiences and lessons from the deployment of our innovative web-based algorithm.

I haven't seen in the whole paper a description of how this solution would be applied into the stream processing of web applications or stream processing. And I don't know where the term "web-based" algorithm comes from. The authors define a ML algorithm, it is not a web-based algorithm. Moreover, the authors have deployed their algorithm in a cloud system, not in a web system.


3. Previous editions of WWW conference had papers in this topic. This is correct, but again this fact does not justify the fitting of the paper into the Systems and Infrastructure for Web, Mobile, and WoT. One should go to previous editions of the conference, identify the tracks offered in those editions and check which track the papers referred by the authors were submitted to. In this year edition, there might be tracks where this paper might be a good fit to, but not the Systems and Infrastructure for Web, Mobile, and WoT track.


In this respect, I haven't seen in my experience a reviewer of probably hundreds of papers, authors adding a "Relevance Statement" into a paper. This is an indication that authors have serious doubts regarding the fit of the paper in this track and that is why they added this paragraph.
Indeed, this is wrong. If the paper is a good fit for the track, authors should be able to make it clear in the introduction of the paper. In the introduction of the paper authors should have been able to frame their work such that there were no doubts that the paper was a good fit for the track, by defining a problem/context which is obviously in the scope of the track.


Having clarify the scope issue, I would like to acknowledge that my expertise falls in the area of the track and thus I consider myself unable to properly assess the quality or novelty of the paper, since it is far from my area of expertise.

My only consideration with regards of novelty is the fact that this paper focus on univariate time series, while as far as I know the most innovative works in the topic of time series and functional data analysis focuses on multivariate data. This in principle would affect the novelty of the paper. But as I acknowledge already, I am not an expert in the topic and then I cannot guarantee the correctness of this statement.


For the above, my review will stick to those aspects I am able to assess such as structure and clarity of the paper and the evaluation of the solution.


1. As I said before the introduction of the paper fails to frame the paper in the scope of the track.

2. The paper is in general well structured; the description of the proposed solution is fair and there is a likewise fair evaluation of the system.


3. My main comments relates with the evaluation of the system:

	* The authors make the following statement: "Regarding hyperparameters, we conducted a grid search to identify the most effective parameters for different datasets". This I assume works well when using tagged datasets. However, when the solution is implemented in a real system, where there is not a ground truth of the anomalies, how the authors can guarantee the solution is operating in its best configuration? For instance, in the cloud system, they've implemented it, how they know the ~10% improvement they are getting is the best the solution can get?

	* The authors only report F1-score, which is fine. However, this does not allow to Understand where the failures of the solutions come from. Reporting other metrics such as accuracy, precision, FPR or TPR, etc. would be beneficial to Understand the performance of the solution in detail.

	* The evaluation lacks a proper analysis of the computational cost of the proposed solution vs. the state-of-the-art. The selection of the algorithm to use in a real use-case depends not only not the offered performance of the algorithm but its associated cost.

**Questions:**

1- Can the authors provide a credible justification regarding the fit of the paper in the track?

2- Could the authors extend the evaluation to other metrics is addition to F1-score?

3- Could the authors provide an analysis of the computational cost of their solution in comparison with the state of the art?

**Reviewer Confidence:**

1: The reviewer's evaluation is an educated guess

**Scope:**

1: The work is irrelevant to the Web

---

### Official Review · Reviewer_gTj8 · 2023-11-23

**Novelty:** 4
**Technical Quality:** 5

**Review:**

This paper addresses the challenges faced by Variational Autoencoders (VAEs) in capturing both long-periodic heterogeneous patterns and detailed short-periodic trends. They proposes the Frequency-enhanced Conditional Variational Autoencoder (FCVAE), an unsupervised anomaly detection method for univariate time series. They incorporate a "target attention" mechanism designed to extract the most valuable information from the frequency domain for improved short-periodic trend construction.

There are a few weaknesses within this paper. For example, it’s relatively hard to read with experimental detail illustrated in the Introduction without a high-level description. The novelty is not clearly demonstrated. Some part of the methodology is not clear enough. And the evaluation metric is limited compared with other similar research.

**Questions:**

1.	In Section 1 Introduction, what’s the “optimal performance” mentioned in line 92?
2.	In Section 1 Introduction, the statement like “we aim to re-examine the VAE model and improve its effectiveness in anomaly detection” is not strong enough in novelty.
3.	In Section 1 Introduction, it’s hard to read with experimental details shown in the introduction part without high-level explanations. For example, the meaning of “reconstruction error” in effectiveness in anomaly detection.
4.	In Section 2.2 VAEs and CVAEs, it’s better to claim what’s the relationship between FCVAE and these two frameworks.
5.	In Section 3, only data augmentation is demonstrated however missing other preprocessing steps in Figure 3. And the correlation between Figure 3 and Figure 5 is not clearly displayed from the figures.
6.	In Experiment results, only the F1 score is utilized for comparison. How about other metrics like accuracy, precision, recall, and etc? Why choose the F1 score only?
7.	There are some small errors. For example, in Section 2.1 line 244, “Given a UTS data”

**Reviewer Confidence:**

2: The reviewer is willing to defend the evaluation, but it is likely that the reviewer did not understand parts of the paper

**Scope:**

4: The work is relevant to the Web and to the track, and is of broad interest to the community

---

### Official Review · Reviewer_cfBX · 2023-11-24

**Novelty:** 5
**Technical Quality:** 6

**Review:**

**Summary:**

Time series anomaly detection plays a crucial role in practical applications. This paper introduces a novel VAE-based method to address existing limitations in VAE-based anomaly detection.
The authors' approach uniquely considers periodic patterns and detailed trends across different frequencies, employing specialized modules for each.
Their results demonstrate superior performance compared to existing methods.

**Strengths:**

- The paper is well-structured, effectively mapping the three identified challenges to the designs of the LFM, GFM, and attention mechanisms.

- The evaluation section comprehensively discusses these designs, offering valuable insights into their implementation and impact.
Weaknesses:

**Weaknesses:**

The overall improvement seems not significant.

**Questions:**

What is the precision and recall performance of the proposed method? While the F1 score provides a holistic view, precision, and recall are often crucial in specific scenarios. They are also used in [34].

**Reviewer Confidence:**

1: The reviewer's evaluation is an educated guess

**Scope:**

4: The work is relevant to the Web and to the track, and is of broad interest to the community

---

### Decision · Program_Chairs · 2024-01-22

**Decision:**

Accept

**Comment:**

In this paper, the authors propose a novel VAE-based method, FCVAE, for time series anomaly detection, addressing limitations in capturing periodic patterns and detailed trends. Reviewers recognize the value and the comprehensive evaluation of the proposed method, but overall reviewer confidence is very low (lowest in my batch of papers). Therefore, there are major concerns regarding the fit of the paper in the scope of the conference track. Also, some raised concerns arise regarding the perceived significance of improvement and clarity in the methodology.

The authors have been very active during the rebuttal phase, trying to convince the reviewers about the fit within the track scope, while also answering technical questions about the results. Even after many discussions, there is still one reviewer who is absolutely not convinced this contribution is fitting to the track, and still recommends a reject. Specifically, the paper monitors a system that is not a web system -- but a cloud system with a web interface for accessing the information related to the cloud system.